# Glycosylation of Stilbene Compounds by Cultured Plant Cells

**DOI:** 10.3390/molecules25061437

**Published:** 2020-03-22

**Authors:** Kei Shimoda, Naoji Kubota, Daisuke Uesugi, Yusuke Kobayashi, Hatsuyuki Hamada, Hiroki Hamada

**Affiliations:** 1Department of Biomedical Chemistry, Faculty of Medicine, Oita University, 1-1 Hasama-machi, Oita 879-5593, Japan; shimoda@med.oita-u.ac.jp (K.S.); nkubota@med.oita-u.ac.jp (N.K.); 2Department of Life Science, Faculty of Science, Okayama University of Science, 1-1 Ridai-cho, Kita-ku, Okayama 700-0005, Japan; dshangshan10@gmail.com (D.U.); junfuquan853@gmail.com (Y.K.); 3National Institute of Fitness and Sports in Kanoya, 1 Shiromizu-cho, Kagoshima 891-2390, Japan; hatsuhama@yahoo.co.jp

**Keywords:** glycosylation, oxyresveratrol, gnetol, β-glucoside, cultured plant cells, *Phytolacca americana*

## Abstract

Oxyresveratrol and gnetol are naturally occurring stilbene compounds, which have diverse pharmacological activities. The water-insolubility of these compounds limits their further pharmacological exploitation. The glycosylation of bioactive compounds can enhance their water-solubility, physicochemical stability, intestinal absorption, and biological half-life, and improve their bio- and pharmacological properties. Plant cell cultures are ideal systems for propagating rare plants and for studying the biosynthesis of secondary metabolites. Furthermore, the biotransformation of various organic compounds has been investigated as a target in the biotechnological application of plant cell culture systems. Cultured plant cells can glycosylate not only endogenous metabolic intermediates but also xenobiotics. In plants, glycosylation reaction acts for decreasing the toxicity of xenobiotics. There have been a few studies of glycosylation of exogenously administrated stilbene compounds at their 3- and 4′-positions by cultured plant cells of *Ipomoea batatas* and *Strophanthus gratus* so far. However, little attention has been paid to the glycosylation of 2′-hydroxy group of stilbene compounds by cultured plant cells. In this work, it is described that oxyresveratrol (3,5,2′,4′–tetrahydroxystilbene) was transformed to 3-, 2′-, and 4′-β-glucosides of oxyresveratrol by biotransformation with cultured *Phytolacca americana* cells. On the other hand, gnetol (3,5,2′,6′–tetrahydroxystilbene) was converted into 2′-β-glucoside of gnetol by cultured *P. americana* cells. Oxyresveratrol 2′-β-glucoside and gnetol 2′-β-glucoside are two new compounds. This paper reports, for the first time, the glycosylation of stilbene compounds at their 2′-position by cultured plant cells.

## 1. Introduction

Oxyresveratrol (3,5,2′,4′–tetrahydroxystilbene) and gnetol (3,5,2′,6′–tetrahydroxystilbene) are the major bioactive stilbenoids. These compounds are contained in plants such as *Artocarpus lacucha* Buch.–Ham and *Gnetum gnemon*, which have been used as folk medicines [1]. They have many pharmacological activities such as anti-oxidant, anti–inflammatory, analgesic, antidiabetic, anti-adipogenesis, anticancer, antileukaemic, and antiviral effects [2,3,4,5,6,7,8,9,10,11,12,13,14,15]. In spite of their pharmacological activities, the water-insolubility of stilbene compounds prevents their further exploitation as medicines and functional food ingredients. Furthermore, these compounds are not necessarily most active within the living body, because of their poor intestinal absorption. The glycosylation of bioactive compounds can enhance their water-solubility and absorption after oral administration [16,17,18].

Plant cell cultures are ideal systems for propagating rare plants and for studying the biosynthesis of secondary metabolites such as flavors, pigments, and agrochemicals, except for a very limited number of compounds (e.g., pyrethrins, bialaphos, and nicotin). Furthermore, the biotransformation of various organic compounds has been investigated as a target in the biotechnological application of plant cell culture systems [19]. Plant cultured cells can be used to convert organic molecules to more useful compounds by catalyzing hydrolysis, oxidation, reduction, esterification, isomerization, and glycosylation reactions. The glycosylation reaction occurs readily in plant cells, because glucosyltransferases are widespread in plants. It has functions such as increasing solubility and stability of aglycons and activating biosynthetic intermediates. Furthermore, it is well known that higher plants accumulate glycosides as secondary metabolites. They are capable of glycosylating not only endogenous metabolic intermediates but also xenobiotics. In plants, glycosylation reaction acts for decreasing the toxicity of xenobiotics. Many studies on the glucosylation of exogenous phenols in cultured plant cells have been reported so far [20,21,22,23,24,25,26,27,28,29]. There have been a few reports describing biocatalytic glycosylation of stilbene compounds [26,27,29]. For example, resveratrol (3,5,4′–trihydroxystilbene) was glucosylated to its 3- and 4′-β-glucosides by cultured plant cells of *Ipomoea batatas* and *Strophanthus gratus* [26]. In addition, pterostilbene (4′–hydroxy-3,5-dimethoxystilbene) and pinostilbene (3,4′–dihydroxy-5-methoxystilbene) were converted into their 3- and/or 4′-β-glucosides by using cultured *Phytolacca americana* cells or glucosyltransferase from *P. americana* [27,29]. These studies reported the glycosylation of stilbenoids at their 3- and/or 4′-positions by cultured plant cells or plant glucosyltransferase. However, little attention has been paid to the biocatalytic glycosylation of stilbene compounds at their 2′-position. These encouraged us to investigate biocatalytic glycosylation of stilbene compounds, which possess a hydroxy group at their 2′-position.

We report here the biotransformations of 2′-hydroxy stilbene compounds, oxyresveratrol and gnetol (Figure 1), by cultured plant cells of *P. americana*. The cultured *P. americana* cells glycosylated these stilbene compounds to oxyresveratrol 3-β-glucoside, oxyresveratrol 2′-β-glucoside, oxyresveratrol 4′-β-glucoside, and gnetol 2′-β-glucoside (Figure 1), in which oxyresveratrol 2′-β-glucoside and gnetol 2′-β-glucoside are two new compounds.

## 2. Results

### 2.1. Glycosylation of Oxyresveratrol (*1*) and Gnetol (*2*) by Cultured Plant Cells of Phytolacca americana

Oxyresveratrol (**1**) was subjected to a biotransformation system using cultured plant cells of *Phytolacca americana*. The compound **1** was administered to flasks containing suspension cell cultures of *P. americana* and incubated at 25 °C for two days on a rotary shaker, after which three products formed (**3**–**5**) were purified by HPLC from the MeOH extract of the cells. Based on the analyses of mass and nuclear magnetic resonance (NMR) analyses including ^1^H and ^13^C-NMR, and HMBC spectroscopy, the chemical structures of the products were determined to be oxyresveratrol 3-β-glucoside (**3**, 36% yield), oxyresveratrol 2′-β-glucoside (**4**, 30% yield), and oxyresveratrol 4′-β-glucoside (**5**, 22% yield). Two products, i.e., oxyresveratrol 3-β-glucoside (**3**) [30] and oxyresveratrol 4′-β-glucoside (**5**) [31], were known compounds. Oxyresveratrol 2′-β-glucoside (**4**) has not been identified before.

Next, gnetol (**2**) was subjected to the same biotransfomation system as described above. After two days’ incubation of cultured cells of *P. americana* with gnetol (**2**), the product **6** was isolated from the MeOH extract of cultured *P. americana* cells. The chemical structure of compound **6** was identified as gnetol 2′-β-glucoside (**6**, 87% yield) based on the analyses of mass and nuclear magnetic resonance (NMR) spectroscopy, i.e., ^1^H and ^13^C-NMR, and HMBC spectra. Gnetol 2′-β-glucoside (**6**) is a new compound.

### 2.2. Determination of the Chemical Structures of New Compounds, Oxyresveratrol 2′-β-Glucoside (*4*) and Gnetol 2′-β-Glucoside (6)

The ESIMS spectrum of **4** showed a pseudomolecular ion [M − H]^−^ peak at *m*/*z* (405) (Appendix A), consistent with a molecular formula of C_20_H_22_O_9_ (calcd. 405 for C_20_H_22_O_9_). The ^1^H-NMR spectrum of **4** had a signal at δ 4.74 (1H, *d*, *J* = 6.8 Hz) corresponding to its attachment to the anomeric carbon (C-1″) (Table 1). The ^13^C-NMR spectrum of **4** exhibited the anomeric carbon signal at δ 100.7 (Table 1). This ^13^C chemical shift of the anomeric carbon at δ 100.7 indicates the presence of *O*-glucoside in the structure of **4** [32,33,34]. From the coupling pattern of the proton signals and the chemical shifts of the carbon resonances due to the sugar moiety (Table 1), the sugar component in **4** was determined to be β-D-glucopyranose. Hydrolysis of **4** using β-glucosidase gave oxyresveratrol (**1**) as the product. This finding shows that the product has *O*-β-glucosylation moiety. The HMBC correlation was observed between the anomeric proton signal at δ 4.74 (H-1″) and the carbon signal at δ 155.6 (C-2′) (Figure 2) to establish that the glucopyranosyl residue was attached to the 2′-hydroxy group of **1**. Thus, the structure of **4** was determined to be oxyresveratrol 2′-β-glucoside.

The ESIMS spectrum of **6** included a pseudomolecular ion [M − H]^−^ peak at *m*/*z* (405), indicating that the product consisted of one substrate and one hexose. The sugar component in the product was determined to be glucose on the basis of the chemical shifts of its carbon signals (Table 1) [34]. The ^1^H-NMR spectrum showed a proton signal at δ 4.89 (1H, *d*, *J* = 7.6 Hz) (Table 1), indicating that the glucoside linkage in the compound had β-orientation. The HMBC spectrum of **6** included the correlation between the proton signal at δ 4.89 (H-1″) and the carbon signal at δ 156.4 (C-2′) (Figure 2). These data indicate that **6** was β-glucosyl analogue of **2**, the sugar moiety of which was attached to the 2′-position of **2**. Thus, the structure of **6** was determined to be gnetol 2′-β-glucoside.

## 3. Discussion

There have been many reports on glucosylation of exogenously supplied simple phenols and polyphenols by cultured plant cells so far [20,21,22,23,24,25]. In most cases, exogenously administered phenols and polyphenols were shown to be converted into their monoglucosides. For example, *Datura innoxia* grown in suspension cultures glucosylated simple phenols, including hydroquinone, resorcinol, and catechol, to their corresponding β-glucosides [20]. Cultured suspension cells of *Gardenia jasminoides* and *Lithospermum erythrorhizon* could produce salicylic alcohol 1-*O*-β-glucoside (salicin) from salicylic alcohol [21]. Suspension cultures of *Datura*, *Lithospermum*, *Perilla*, and *Catharanthus* were capable of glucosylating esculetin to esculetin 7-*O*-β-glucoside (esculin) [22]. A root culture of *Panax ginseng* was able to convert 3,5-dimethoxyphenol (taxicatigenin) into 3,5-dimethoxyphenyl 1-*O*-β-glucoside (taxicatin) [23]. Grapefruit (*Citrus paradisi*) cells in suspension cultures were able to glucosylate exogenous naringenin and hesperein to naringenin 7-*O*-β-glucoside (naringin) and hesperein 7-*O*-β-glucoside (hesperidin) [24]. Cultured cells of *Eucalyptus perriniana* converted daidzein to daidzein 7-*O*-β-glucoside (daidzin) [25].

We recently reported the glycosylation of stilbene compounds by cultured plant cells [26,27]. Resveratrol (3,5,4′-trihydroxystilbene) was glucosylated to give its β-glucosides, i.e., resveratrol 3-*O*-β-glucoside and resveratrol 4′-*O*-β-glucoside, by cultured plant cells of *I. batatas*, *S. gratus*, and *P. americana* [26,27]. It has been reported that piceatannol (3,5,3′,4′-tetrahydroxystilbene) was converted into piceatannol 4′-*O*-β-glucoside by cultured *P. americana* cells [27]. Furthermore, piceatannnol was subjected to the bioconversion with glucosyltransferase from *P. americana* expressed in *Escherichia coli* [28]. The enzyme regioselectively glucosylated piceatannnol at its 4′-position to piceatannol 4′-*O*-β-glucoside. Neither piceatannol 3-*O*-β-glucoside nor piceatannol 3′-*O*-β-glucoside were obtained. These demonstrated that cultured plant cells and plant glucosyltransferase can glucosylate stilbene compounds at their 3- and 4′-positions to the corresponding monoglucosides. No studies on glycosylation of stilbene compounds at their 2′-position by biocatalysts have been reported so far.

The present study describes the glucosylation of stilbene compounds using cultured plant cells of *P. americana* as biocatalysts. When stilbene compounds with hydroxy groups at their 3-, 2′-, and/or 4′-positions were used as substrates, cultured *P. americana* cells introduced glucosyl residue at these hydroxy groups. Cultured *P. americana* cells converted oxyresveratrol, which has hydroxy groups at its 3-, 2′- and 4′-positions, into 3-, 2′-, and 4′-β-glucosides of oxyresveratrol. Gnetol with 3- and 2′-hydroxy groups was transformed to 2′-β-glucoside of gnetol by biotransformation with cultured *P. americana* cells. Gnetol 3-β-glucoside was not formed. This might be explained by substrate specificity of the glucosyltransferases in *P. americana* cells. These results show that cultured cells of *P. americana* can convert exogenous stilbene compounds into their monoglucosides. This paper reports, for the first time, the biocatalytic glycosylation of stilbene compounds at their 2′-position by cultured plant cells.

Oxyresveratrol has been isolated from the roots of *Morus nigra* [35] and from the fruits of *Artocarpus heterophyllus* [36]. A phenolic glycoside was isolated from *Morus lhou* [37] and *Morus alba* [38], and was identified as oxyresveratrol 3,4′-β-diglucoside. It was found that oxyresveratrol 2′-β-glucoside is not biosynthesized in these plants. On the other hand, gnetol was isolated from stems of *Gnetum klossii* [39]. Gnetol was found to exist as an aglycon in *Gnetum gnemon* products [13]. However, gnetol 2′-β-glucoside was not isolated from these plants. In the present study, cultured plant cells of *P. americana* were capable of transforming oxyresveratrol and gnetol to their 2′-β-glucosides. These suggest that glucosyltransferases, which can glucosylate the 2′-hydroxy group of oxyresveratrol or gnetol, were not included in *A. heterophyllus*, *M. lhou*, *M. alba*, *G. klossii*, and *G. gnemon*, but in *P. americana*.

Despite pharmaceutical properties of stilbene compounds, they have shortcomings such as low solubility in water and poor absorption after oral administration. These shortcomings prevent stilbene compounds from being used as drugs and functional food ingredients. Glycosylation is a characteristic reaction which converts water-insoluble and unstable aromatic compounds into the corresponding water-soluble and stable compounds. It has been reported that curcumin, which was supplied exogenously, was converted to its glycosides with drastically increased water solubility by *Catharanthus roseus* cell suspension cultures [16]. Cultured *C. roseus* cells transformed curcumin to various glucosides not only by introducing individual glucose residues onto its two phenolic hydroxy groups but also by forming β-1,6-glycosidic linkages, leading to the gentiobioside derivatives. Glucosyl conjugation of curcumin drastically enhanced the water-solubility. Curcumin 4′-*O*-glucoside and curcumin 4′-*O*-gentiobioside were 230-fold and 2,400,000-fold more soluble than curcumin [16]. On the other hand, it has been reported that the glycosylation of the flavonoid aglycon influences its absorption [17,18]. In the case of quercetin glucoside, the efficiency of absorption was higher than that for quercetin itself [17,18]. Hollman et al. elucidated the mechanism by which glucosylation facilitates quercetin absorption: the glucosides could be transported into enterocytes by the glucose transporter SGLT1 and could then be hydrolyzed inside the cells by cytosolic β-glucosidase [17]. These studies indicate that the glycosylation of bioactive compounds can enhance their water-solubility and intestinal absorption. The glucosides of stilbenes obtained here would be expected as water-soluble food ingredients and drugs. Studies on the pharmaceutical properties of the stilbene glycosides are now in progress in our laboratory.

## 4. Materials and Methods

### 4.1. General

The substrates, i.e., oxyresveratrol and gnetol, were purchased from Sigma Aldrich Japan Inc., Tokyo, Japan.

### 4.2. Analyses

The structures of products were determined based on the analysis of ESIMS, ^1^H- and ^13^C-NMR, and HMBC spectra. The ^1^H- and ^13^C-NMR, and HMBC spectra were recorded using a JNM-ECS400 spectrometer (JEOL Ltd., Tokyo, Japan) in DMSO-*d*_6_ solutions, and chemical shifts are expressed in δ (ppm) with reference to TMS. The ESIMS spectra were measured at 70 eV in negative mode using a JMS-700 MStation (JEOL) in CH_3_OH solution. Products were analyzed by high-performance liquid chromatography (HPLC) using CrestPak C18S column (column size: 150 × 4.6 mm) at column temperature of 40 °C.

### 4.3. Cultivation of Plant Callus

Cultured plant cells of *P. americana* were sub-cultured at 4-week intervals on solid medium containing 2% glucose, 1 ppm 2,4-dichlorophenoxyacetic acid, and 1% agar (adjusted to pH 5.7) in the dark. A suspension culture was started by transferring 20 g of the cultured cells to 300 mL of liquid MS medium in a 500 mL-conical flask. 

### 4.4. Glycosylation by Cultured Plant Cells

The substrate was transformed by using plant cultured cells of *P. americana* as biocatalysts. The cultured *P. americana* cells in the stationary growth phase have been used for experiments. To a 300 mL flask containing 100 mL of the culture medium and suspension cultured cells of *P. americana* (25 g) was added 15 mg of substrate, i.e., oxyresveratrol and gnetol. The culture was incubated at 25 °C for 2 days on a rotary shaker (120 rpm). After the incubation period, the cells and medium were separated by filtration with suction. The filtered medium was extracted with ethyl acetate (AcOEt). The cells were extracted by homogenization with MeOH, and the resulting extract was concentrated. The residue was partitioned between H_2_O and AcOEt. The AcOEt layer was evaporated and the residue was re-dissolved in MeOH and purified by preparative high-performance liquid chromatography (HPLC) [column: CrestPak C18S (150 × 4.6 mm); flow rate: 1.0 mL/min; column temperature: 40 °C; solvent: CH_3_CN:H_2_O = 14:86; retention time (min) of the products, oxyresveratrol 3-β-glucoside (8.7), oxyresveratrol 2′-β-glucoside (12.2), oxyresveratrol 4′-β-glucoside (7.2), and gnetol 2′-β-glucoside (15.6)].

## 5. Conclusions

The present study demonstrated the biotransformation of stilbene compounds that have hydroxy groups at 3-, 2′-, and/or 4′-position by cultured plant cells. It was found that oxyresveratrol and gnetol were glycosylated to 3-, 2′-, and 4′-β-glucosides of oxyresveratrol, and 2′-β-glucoside of gnetol by cultured cells of *P. americana*. Oxyresveratrol 2′-β-glucoside and gnetol 2′-β-glucoside are two new compounds. This paper reports, for the first time, the biocatalytic glycosylation of stilbene compounds at their 2′-position by cultured plant cells.

## Figures and Tables

**Figure 1 molecules-25-01437-f001:**
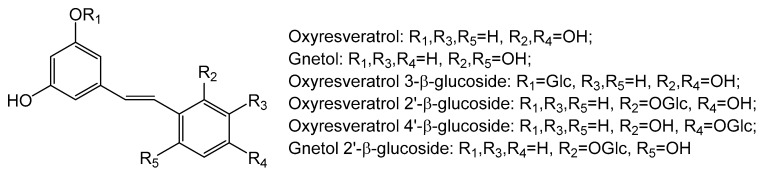
Chemical structures of stilbene compounds.

**Figure 2 molecules-25-01437-f002:**
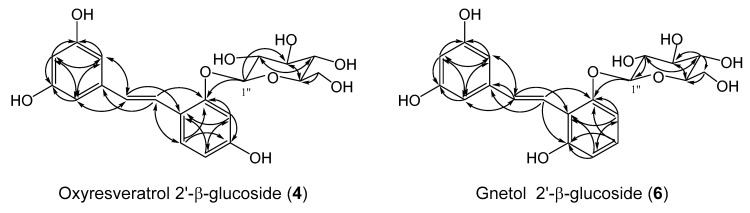
HMBC correlations of oxyresveratrol 2′-β-glucoside (**4**) and gnetol 2′-β-glucoside (**6**).

**Table 1 molecules-25-01437-t001:** ^1^H and ^13^C-NMR data of oxyresveratrol 2′-β-glucoside (**4**) and gnetol 2′-β-glucoside (**6**).

	Position	4		6	
		δ_C_	δ_H_ (*J* in Hz)	δ_C_	δ_H_ (*J* in Hz)
Aglycon	1	139.7		140.8	
	2	104.3	6.34 d (1.6)	104.3	6.38 d (2.0)
	3	158.3		158.4	
	4	101.4	6.03 s	101.6	6.10 t (1.6)
	5	158.3		158.4	
	6	104.3	6.34 d (1.6)	104.3	6.38 d (2.0)
	7	125.7	6.74 d (16)	131.7	7.46 d (17.2)
	8	122.5	7.26 d (16.4)	119.9	7.35 d (16.4)
	1′	117.6		113.5	
	2′	155.6		156.4	
	3′	102.9	6.51 d (1.6)	106.0	6.57 d (8.4)
	4′	158.1		127.8	6.97 t (8.4)
	5′	109.4	6.40 dd (8.8, 1.6)	109.6	6.64 d (8.4)
	6′	126.5	7.40 d (8.4)	156.5	
Glc	1″	100.7	4.74 d (6.8)	100.6	4.89 d (7.6)
	2″	73.3	3.44 m	73.6	3.30 m
	3″	76.5	3.26 m	77.0	3.19 t (8.8)
	4″	69.5	3.26 m	69.7	3.31 m
	5″	76.9	3.18 m	77.1	3.38 m
	6″	60.5	3.45 m,	60.7	3.69 d (10.8),
			3.73 m		3.47 dd (12.0, 5.6)

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
