# Peer review of "Glycosylation of Stilbene Compounds by Cultured Plant Cells"

_molecules, 2020, doi:10.3390/molecules25061437_

Round 1

Reviewer 1 Report

Authors have followed the suggestions previously reported. In my opinion the the article has been improved and it can be accepted for its publication in Molecules

Author Response

Dear sir/madam,

Thank you very much for everything.

With best regards,

Kei Shimoda

Reviewer 2 Report

This paper presents convincing data on the conversion of oxyveratrol and gnetol into beta-glucuronides by Phytolacca americana cells cultures.

The NMR data are correctly explained and convincing. The Mass spectra could be nicer. PLese explain they look like MS2 spectra, not ESI-MS spectra where the molecular ion (M+H+) is a major peak.

However in the description of the experiment there are strange description of the analytical and semi-preparative column.

Did the actual experimenter write the manuscript. Please check and correct (or explain)

----------------------------

There is something wrong in the column and flow rate description.
On a 30 mm ID column the flow rate is at least 10 and more 30 ml/min.
1 mL/min would correspond to an analytical column : 3.0 to 4.6 mm ID.

By looking in the Jasco catalog the reference Crestpak C18 S corresponds to clumn 4.6 mmm ID 150 mm length.

There is also semi-preparative column UNIPAK 30 mm ID, A50 mm long.

Please precise correctly the column used. and flow rate in analytical and semi-preparative HPLC.

Thus I suggest tou fix this important point. and comment your mass spectra.

Then the paper would be acceptable for publication

Author Response

Dear sir/madam,

Thank you very much for your comments.

The molecular weight of both oxyresveratrol 2’-glucoside and gnetol 2’-glucoside is 406 (C20H22O9).

406-1=405

Their pseudomolecular ion peak was observed at m/z 405.

So, the ESIMS data of these compounds were measured in [M-H]- mode.

The column size has been corrected to 150×4.6 mm(, which can be recognized by yellow background).

With best regards,

Kei Shimoda

Reviewer 3 Report

The revised manuscript can now be accepted

Author Response

Dear sir/madam,

Thank you very much for everything.

With best regards,

Kei Shimoda

This manuscript is a resubmission of an earlier submission. The following is a list of the peer review reports and author responses from that submission.

Round 1

Reviewer 1 Report

The manuscript “Glycosylation of Stilbene Compounds by Cultured Plant Cells” is of interest and can be accepted if the authors provide more information.

The authors should add all spectra to the SI. All the MS and the NMR data should be there so see and compare so that the information in Fig. 2 and table 1 are clear and comparable.

Small comments: The ESI MS mass is normally measure in positive mode and then a proton is taken up. The mass is plus 1 and thus uneven. The authors should give full MS description and either explain why negative mode was used or correct the minus charge.

In table 1 agricon should be aglycon

In the materials and methods part add the ESI MS details and also add the size of the HPLC column.

Reviewer 2 Report

Comments on the manuscript ”molecules-669332”

Specific comments

The language needs to be checked by a native English speaker. It is not in acceptable state for publication. For example, on line 13 in the Abstract, the word ”being” needs to be inserted (twice, in fact). The Abstract, in particular, needs a lot of revision of the language. There are even misspelled words: for example, ”P. americna” in more than one place in the manuscript. There are several places in the manuscript, which have English language mistakes.

The Abstract is not weighted well. It has far too much background information (theory) in the beginning – some can be deleted, since it is not directly relevant to the focus of the study. Unfortunately, the actual methods and results, and their significance are mentioned too briefly. So, these parts need more coverage in the Abstract. There is no hypothesis mentioned: Why would plant cell cultures (of Phytolacca americana) produce a glycosylation effect? Was glycosylation known previously from other plant systems? Please mention this in the abstract. This is also an important aspect which is missing in the Introduction.

There is too much on the various pharmacological properties and mode of action, both in the Introduction and the Abstract, when this is not the focus of the study in the manuscript. Pharmacological properties and effect in the human body are not tested in this study, nor is mode of action. Instead, the Abstract (and the Introduction and Discussion) should have more emphasis on the actual focus of the study – the glycosylation reaction(s). Are they present in other systems? Please present other scientific studies and place your work/your results in relation to them.

Line 93 (Introduction) mentions that plant cells are excellent for studying these systems, but no reference is made to the literature. This would be an excellent place to mention other glycosides produced in plant cell cultures, with reference to the literature. You already have some references (for example no. 28), but you need to read it and other papers like it carefully, and refer to the results of these papers individually (not as a group of references, where no. 28 is buried now).

The Discussion is too thin (too short and too few references). Probably for the same reason, i.e. that studies similar to the ones presented have not been mentioned – there is no clear hypothesis in the Introduction (for example that would answer why would you expect a glycosylation effected by plant cell cultures?). Since no similar studies are presented in the Introduction, it becomes difficult to relate the results to similar results of others in the Discussion. More writing needs to be done on this aspect by the authors both in the Introduction and the Discussion.

In lines 189-192: “There have been several reports that describe biotransformation of stilbene compounds by biocatalysts such as cultured plant cells and plant glucosyltransferases. It has been reported that cultured cells of P. americana converted resveratrol with hydroxy groups at its 3- and 4'-positions into the corresponding  3- and 4'-β-glucosides.”

These sentences should be followed by literature references, especially the one claiming that: “there are several reports”. There should be a better description as to which of these are close to your study and if so, in what way, and in which way they are different to your study?

Conclusion (line 250): “This is the first report describing the biocatalytic glycosylation of stilbene compounds at their 2'-positions.” If it is the first report (which is often hard to prove, because it is difficult to cover all published literature in literature searches), and it is only the 2’ position that is new, where are other studies presented in this manuscript that have described the biocatalytic glycosylation of stilbene compounds at other positions?

Methods and Results. The only comment is on the spectra presented in the supplementary material (figures), where the quality (resolution) could be better, and where the structural formula of the compounds that are studied/sought need to be presented next to the peaks in the NMR spectra, as is customary.

Reviewer 3 Report

Shimoda et al. resport the glycosylation of different stilbene compounds mediated by cultured plant cells of Phytolacca americana. Glycosylation of compounds is a common strategy to improve the pharmacokinetic profile of bioactive molecules and, as it is described in this work, it could be applied to stilbene derivatives. In addition, the employ of cultured plant cells enhances the sostainability of the process. However, there are some important points that which should be considerably improved.

English languaje should be widely revised. As some examples: 

page 1, line 41, 'Recently, a pharmacokinetics of oxyresveratrol in rats by oral dosing have been examined' must be corrected: 'recently, a pharmacokinetic study of...HAS been examined' page 2, line 49, 'Gnetum gnemon products HAVE been reported instead of has been. Page 2, lines 49-51 should be re-written....

The abstract should be more concise, and it should be included a sentece to introduce the work and the novelty of the study. For example, in line 23, it could be included 'In this work it is described the transformation of oxyresveratrol to...'; other way, it is not clear the contribution of authors.

The Introduction section  should be considerably reduced. There are many data which should be deleted as they do not belong to the main topic of the work. Pharmacological effects of gnetol could be reduced, or the effect of piperine (lines 84-92) could be deleted, as well as the effect of esters of caffeic acid, among others.

Authors describe the glucosylation of different compounds on different positions. However, no data about the HPLC retention times (the composition of the mobile phase is missed) are reported. They describe the obtention of different compounds which are identified by NMR, but no data is included in the work. At least H1-NMR data should be included for compouns 5, 6, 10, 11, 12 and 13. They describe the NMR data for compounds 4 and 7, which are the novelty of the work. However in the supporting information the spectra show more signals (products are not completely pure). It should be also included the HMBC spectra in the SI. Please, indicate which spectra corresponds to each compound in the SI.

The compounds on Figure 2 do not contain a glucose molecule but a C-4 epimer.

As it is reported above, different important points should be improved, changed and corrected before considering the publication of this work in Molecules.

Reviewer 4 Report

This paper present a study on glycosylation of oxy-veratrol by plant cells

The paper is well written and clear.

It would have been interessant for some readers like be to give more informetions in the supplement part on HPLC traces and  physical data of the isolated compounds.

The NMR analysis seem well done and clearly prove the structures of the discussed compounds.

I find this paper interesting for specialists and believe it should be published.

I notted a few spelling errors which should be corrected.